# Variations in Ozone Concentration over the Mid-Latitude Region Revealed by Ozonesonde Observations in Pohang, South Korea

**Daegeun Shin, Seungjoo Song, Sang-Boom Ryoo and Sang-Sam Lee ***

Innovative Meteorological Research Department, National Institute of Meteorological Sciences, 33 Seohobuk-ro, Seogwipo-si, Jeju-do 63568, Korea; shingeun@korea.kr (D.S.); sjtomato@korea.kr (S.S.); sbryoo@korea.kr (S.-B.R.)
* Correspondence: sangsam.lee@korea.kr

**Abstract:** Ozone absorbs harmful UV rays at high elevations but acts as a pollutant gas in the lower atmosphere. It is necessary to monitor both the vertical profile and the total column ozone. In this study, variations in the ozone concentration of Pohang were divided into three vertical layers: the stratospheric layer (STL), the second ozone peak layer (SOPL), and the tropospheric layer (TRL). Our results indicated that the ozone concentration in the STL, SOPL, TRL, and total column ozone increased by 0.45%, 2.64%, 5.26%, and 1.07% decade$^{-1}$, respectively. The increase in the SOPL during springtime indicates that stratosphere–troposphere exchange is accelerating, while the increase during summertime appears to have been influenced by the lower layers. The growth of tropospheric ozone concentration is the result of both increased ozone precursors from industrialization in East Asia and the influx of stratospheric ozone. Our results reaffirmed the trend of ozone concentration in mid-latitudes of the northern hemisphere from vertical profiles in Pohang and, in particular, suggests that the recent changes of ozone in this region need to be carefully monitored.

**Keywords:** ozone; second ozone peak; stratosphere; troposphere; stratosphere-troposphere exchange

## 1. Introduction

Ozone is a gas that plays a crucial role in the thermal and chemical balance of Earth's atmosphere. Stratospheric ozone, which accounts for approximately 90% of the earth's atmospheric ozone, absorbs the sun's harmful UV rays, thereby protecting humans and ecosystems. Variations in stratospheric ozone concentration have an impact on surface ozone concentration and, moreover, the future climate [1]. The amount of stratospheric ozone declined sharply due to the indiscriminate use of ozone-depleting substances containing halogens until the Montreal Protocol was adopted in 1987. The rate of reduction in the stratospheric ozone (around 42 km altitude) reached 8% per decade from 1970 to 1997 [2]. The Montreal Protocol implemented strong measures to limit the use of ozone-depleting substances globally, and many studies have reported that stratospheric column ozone began increasing again after the mid-1990s [3–10]. However, recent reports indicate that the global emission of CFC-11, an ozone-depleting compound, has increased in recent years [11–13], especially in eastern mainland China (provinces of Anhui, Beijing, Hebei, Jiangsu, Liaoning, Shandong, Shanghai, Tianjin, and Zhejiang) [11]. Therefore, the ozone concentration downwind of China needs to be closely monitored to investigate the effect of increased CFC-11 emissions on ozone concentration.

Studies have reported that the frequency of stratospheric–troposphere exchange (STE) of ozone increases as the total ozone increases [14–19]. Because the concentration of ozone in the stratosphere is much higher than that in the troposphere, the STE of ozone generates a second ozone peak between the upper troposphere and the lower stratosphere [16]. This second ozone peak causes a sharp increase

in the ozone concentration in the vicinity of the tropopause, which not only increases the total ozone [14], but also often affects the surface ozone concentration [20]. Recent studies have shown that the STE is accelerating due to climate change [14–19], the direct cause of which is reported to be the increased blocking of synoptic waves due to global warming [15,17,21,22]. The concentration of tropospheric ozone also shows an obvious increasing trend caused by various factors, such as an increase in precursors, transport of ozone from surrounding areas, and changes in regional climate characteristics [2,15,23].

The formation and destruction of ozone varies with its altitude; therefore, observations must be made at multiple altitudes to understand ozone variation in the atmosphere. The Korea Meteorological Administration (KMA) has performed vertical ozone observations in Pohang since 1995. Pohang is a coastal city situated on the eastern coast of the Eurasian continent and sits to the leeward side of China. It is located in the mid-latitude region and is influenced by westerlies. East Asia, which has undergone rapid industrialization in recent years, is one of the regions drawing significant attention for its contribution to global climate and environmental changes. Therefore, long-term vertical ozone data accumulated in Pohang is effective for monitoring and studying changes in ozone concentration over the mid-latitude region of the northern hemisphere. In this study, the characteristics of ozone variations in the mid-latitude region due to recent rapid industrialization, in addition to global climate change, were investigated using vertical ozone data from Pohang obtained from ozonesonde. The vertical ozone concentration was divided into three layers according to altitude—the stratospheric layer (STL), the second ozone peak layer (SOPL), and the tropospheric layer (TRL)—and the variation trends in each layer were analyzed. Furthermore, the causes of ozone variations were investigated based on the variation pattern of the associated meteorological components.

## 2. Materials and Methods

### 2.1. Data and Period

In this study, vertical ozone concentrations in Pohang were measured using ozonesonde. These measurements were used to analyze trends in ozone variations in the mid-latitude region for a 21-year period, from January 1997 to December 2017. Additionally, various meteorological and surface ozone data were used to analyze the causes of variations in the ozone concentrations. For the humidity data, observation values from the radiosonde attached to the ozonesonde were used, and for the wind direction and wind speed, the rawinsonde data collected twice daily (at 00:00 UTC, 12:00 UTC) were used. Lastly, for the surface ozone, data from the three air monitoring stations in Pohang (Jangheung-dong, Jukdo-dong, and Daedo-dong) operated by the Korean Ministry of Environment, were obtained for the 2001 to 2017 period. Surface ozone concentrations were observed hourly, and the mean value of the data obtained from the three stations was used for trend analysis. Trend analysis was carried out using monthly averaged data for the whole study period (1997 to 2017), except for surface ozone (2001 to 2017). A backward trajectory analysis was performed using the NOAA's Hybrid Single Particle Lagrangian Integrated Trajectory Model (HYSPLIT) trajectory model and Global Data Assimilation System (GDAS) meteorological data.

### 2.2. Electrochemical Concentration Cells Ozonesonde

There are three common types of ozonesonde presently in use: the Brewer-Mast [24], Electrochemical Concentration Cells (ECC) [25], and carbon iodine cell [26]. In the Pohang meteorological station, an ECC-type ozonesonde is used. This type of ozonesonde is used by more than 80% of the worldwide WMO/ Global Atmosphere Watch (GAW) ozone sounding network due to its relative stability [27]. The ECC ozonesonde measures the current flow caused by the reaction of ozone and the potassium iodide (KI) solution inside the reaction cell. The amount of ozone is determined by the intensity of the reactive current [27–29]. The ECC ozonesonde has been reported to have a precision of 3–5% and an accuracy of approximately 5–10% at most altitudes below

28 km [27,30,31]. The ozonesonde observations in Pohang are conducted once a week, at approximately 05:00 UTC on a clear day.

*2.3. Data Selection*

To obtain the total column ozone using an ozonesonde, an estimated residual ozone column above the balloon burst altitude is added to the integrated sonde profile. The integrated total ozone column can be adjusted to an independent total ozone column measurement (Brewer ozone spectrophotometer or satellite observation), and the ratio between the two sets of total ozone columns is called the Normalization Factor (NF) [29]. The adjustment using NF can reduce the uncertainty of an integrated ozonesonde profile, particularly in the stratosphere [32]. However, Smit et al. [33] reported that the correction of ozonesonde data using NF conflicts with the fact that the electrochemical ozonesonde, in principle, is an absolute measuring device. Nevertheless, it provides a screening test for unreliable soundings [33,34]. Therefore, in this study, with reference to previous studies, only data corresponding to NF values from 0.8 to 1.2 were used for analysis [29,33,34].

The climatology obtained from the Solar Backscattered Ultraviolet (SBUV) [35] was used as an estimate for the ozone amount above the balloon burst altitude, and for the independent total ozone column measurement, the Brewer ozone spectrophotometer data observed in Pohang was used. In addition, the Ozone Monitoring Instrument (OMI)/Aura satellite data [36] were employed to replace the unstable Brewer ozone spectrophotometer data obtained during the 2012–2014 period from an aging instrument, any missing data for the analyzed time period, and, from 2017 onwards, when observations ceased.

*2.4. Altitude Classification for Analysis*

Ozone in the Earth's atmosphere has different sources at different altitudes. Ozone near the surface is generated through photochemical reactions of nitrogen oxides and volatile organic compounds, which are representative precursors, and it is generated most intensively in early summer when sunlight is strong. Conversely, ozone in the stratosphere is generated by the combination of oxygen molecules and atomic oxygen, separated by strong UV rays. A large amount of ozone generated near the equator with strong incident sunlight undergoes global circulation (Dobson–Brewer circulation) and in the mid-latitude regions of the northern hemisphere (where the Korean Peninsula is located), the highest ozone values are observed in spring. Ozone in the troposphere and stratosphere are vertically connected and are exchanged through the movement of the atmosphere. Therefore, the vertical ozone profiles and the elevation of the tropopause (the boundary between the troposphere and stratosphere) vary seasonally. Observations of the second ozone peak predominately near the tropopause, approximately 9–16 km in Pohang, and this altitude section is very important for examining the ozone variations in the stratosphere and troposphere [14,16]. We aimed to analyze the trend of changes in the ozone columns while carefully examining the STE of ozone in the SOPL. Although the tropopause is usually considered the boundary between the troposphere and stratosphere, we instead define the 9–16 km section where the tropopause and the second ozone peak are located as the SOPL, and sections higher than 16 km and lower than 9 km as the STL and TRL, respectively. The ozone column in the SOPL and the TRL was obtained by vertically integrating the ozonesonde observations. The ozone column in the STL was calculated by subtracting the combined ozone columns in the TRL and SOPL from the independent total column ozone observed by the Brewer ozone spectrophotometer (or satellite) to correct for the uncertainties of sonde observation in the stratosphere.

## 3. Results

*3.1. Trend Analysis*

The trends of ozone column for each atmospheric layer are shown in Figure 1. The amount of ozone in each layer is expressed as a time series of the percentile deviation of the ozone column with

respect to the monthly climatology. Since the ozone concentration has a strong seasonality, the seasonal component was removed by subtracting the mean annual cycle presented in Figure 2. For the trend analysis, a first-order regression analysis was used, and the trend was expressed in the form of decadal variations for each layer in the same manner as in previous studies [2–4,14]. As a result, during the study period, the STL, SOPL, and TRL all showed increasing ozone trends of 0.45%, 2.64%, and 5.26% decade$^{-1}$, respectively. Overall, the ozone column in the STL increased the least. The SOPL and particularly the TRL showed a greater increase. The total ozone also increased by 1.07% decade$^{-1}$, which indicates that the increasing ozone in the SOPL and the TRL contributed greatly. The changes in ozone concentration since the Montreal Protocol have followed a recovery trend, whereas the continuous decrease of ozone concentration observed in the STL during 2011 to 2016 may have been related to an increase in CFC-11 emissions [11–13]. However, due to climate change and other ozone depleting substances, the effect of CFC-11 on ozone concentration is very complex, with a highly variable response time [12,37,38]. Therefore, the data need to be interpreted cautiously. Meanwhile, the increasing ozone trend observed in the SOPL appears to result from accelerated STE. This is because the enhanced ozone intrusion from the stratosphere caused by the enhanced STE increases the ozone concentration of SOPL. In this regard, it is estimated that the ozone increase in TRL was influenced by the ozone intrusion in conjunction with increased surface pollutants.

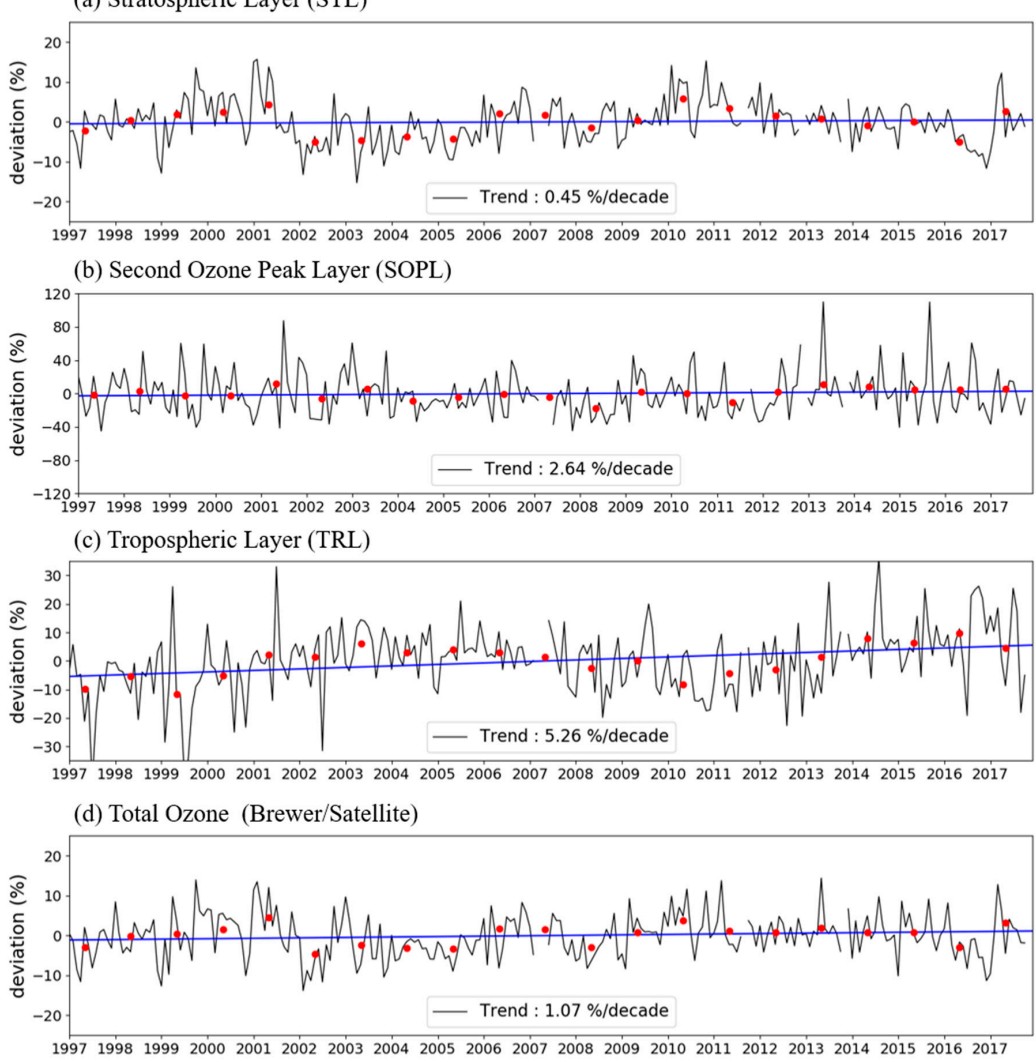

**Figure 1.** Time series of percent deviations in deseasonalized average ozone (DU) for three layers—(**a**) Stratospheric layer, (**b**) Second ozone peak layer, (**c**) Tropospheric layer and (**d**) the total column. The blue line and red dots represent the trend and annual mean of each layered ozone, respectively.

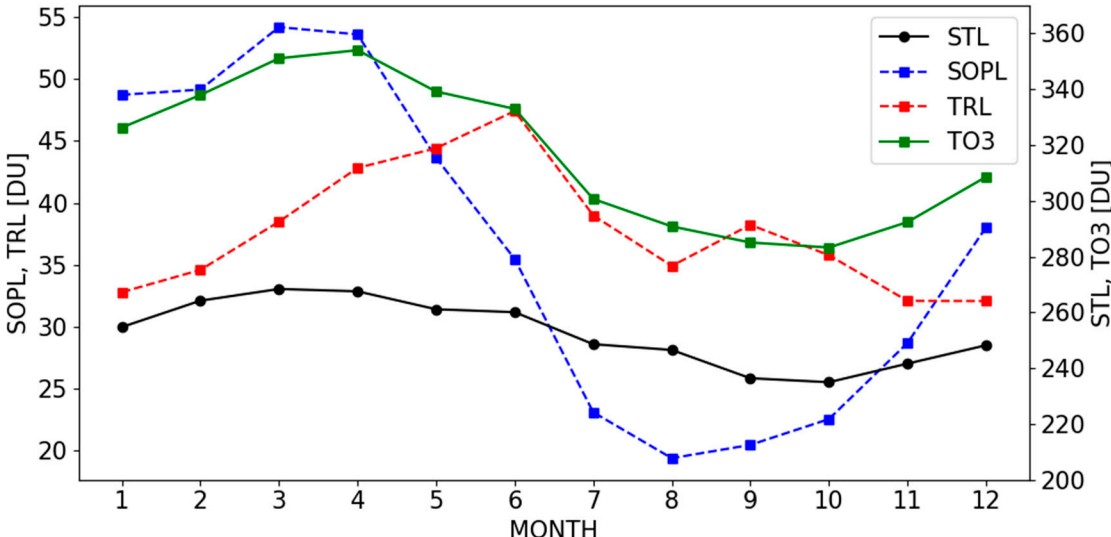

**Figure 2.** Monthly variation of each ozone layer (SOPL: second ozone peak layer, TRL: tropospheric layer, STL: stratospheric layer, TO3: total column ozone).

The seasonal ozone variations for each layer are illustrated in Figure 3. The ozone in the STL shows the highest rate of increase in the spring, while ozone concentrations in both the SOPL and the TRL increase significantly in the summer. In particular, the increase of ozone in summer is highest in the TRL, while a negligible variation in the ozone was observed in the STL. The biggest source of tropospheric ozone is the photochemical reaction of surface precursors, and because summer is characterized by strong sunlight and active vertical mixing, the transport of pollutants from China also greatly impacts the troposphere over the leeward area during this period [39]. Therefore, considering that there was no noticeable increase in radiation during the period, from the analysis of radiation data (not included in this paper), the increase in the TRL ozone is most likely significantly affected by industrialization in East Asia, including China and Korea. In summer, as vertical mixing becomes stronger, the tropopause rises, and the actual troposphere includes the SOPL. Therefore, the ozone increase observed in the SOPL can also be attributed to the increased ozone in the TRL. In spring, the ozone column in the STL shows a relatively strong increase, and the ozone in the SOPL shows a more significant increasing trend than ozone in the TRL. STE is more active in spring [14,40], and the high rate of ozone increase during this period is thought to be mainly due to the strengthening of the second ozone peak due to the influx of stratospheric ozone.

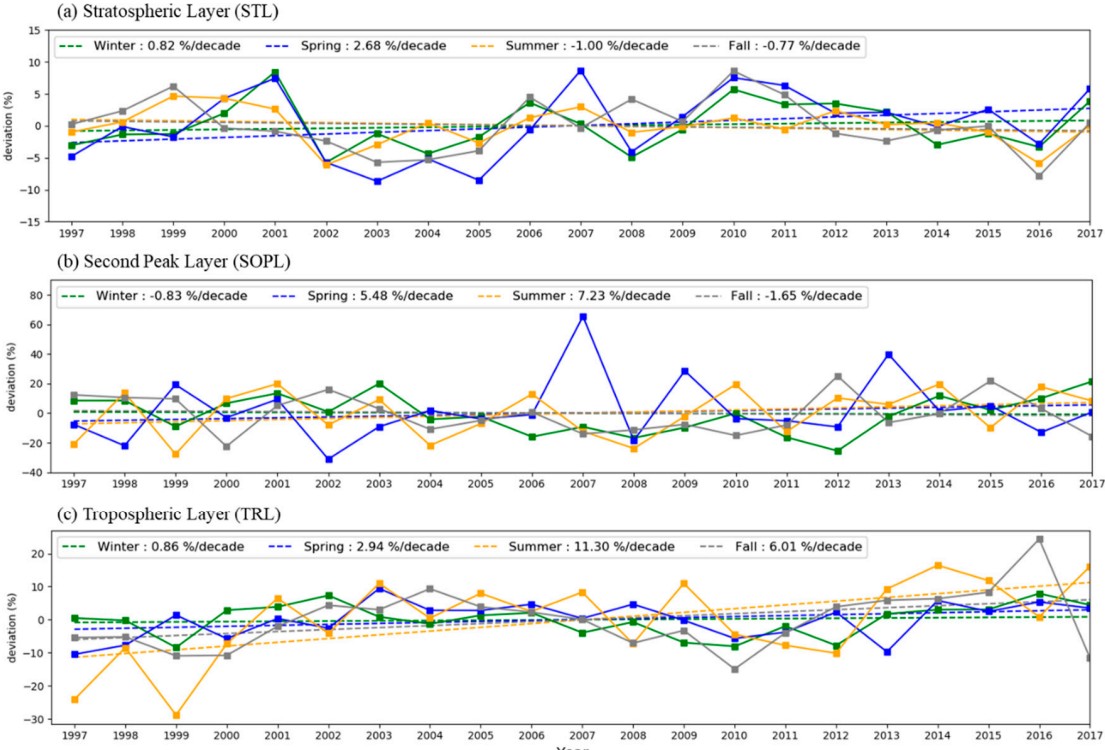

**Figure 3.** Time series of percent deviations in de-seasonalized average ozone (DU) of three layers—(**a**) Stratospheric layer, (**b**) Second ozone peak layer, (**c**) Tropospheric layer—for each season. The solid and dashed lines indicate annual mean and trend, respectively (green: winter, blue: spring, yellow: summer, gray: fall).

## 3.2. Influence Components Analysis

The reasons presented for the observed ozone variations can be investigated in more detail using the variation of the components related to the generation and movement of ozone. We selected four main components which affect ozone concentration: wind speed and vertical wind shear, both of which impact the development of the second ozone peak; relative humidity to determine the effect of the dry air originating from the stratosphere [34]; and surface ozone, which is closely related to tropospheric column ozone [14,16,41]. The vertical wind shear was calculated using the standard deviation of wind speed for each altitude of the corresponding layer, and the TRL was subdivided into the lower troposphere (0–3 km) and the upper troposphere (3–9 km) based on the planetary boundary layer to further clarify the correlation between each component.

Figure 4 represents a correlation heatmap showing the relationship between the selected components and the ozone variations in each layer. The heatmap is shown separately for spring and summer, when the increase of the ozone concentration in the SOPL was significant. When the seasonality was removed, it was difficult to examine the direct relationship between the components, so the correlation analysis was performed based on the monthly mean value without removing the seasonality. In the correlation heatmap, components showing positive correlations are displayed in red, and those with negative correlations are displayed in blue, and the stronger the correlation, the darker the color. Our analysis first focused on ozone concentrations in the SOPL and TRL, which showed significant increases in the previous trend analysis. In the correlation heatmap for spring (Figure 4a), the SOPL ozone showed a strong correlation with both the total ozone and the STL ozone, but a relatively low correlation with the TRL ozone, despite its proximity. This low correlation can be explained by the different seasonal cycles between the two layers, as can be seen in Figure 2. Because the SOPL ozone mainly follows the seasonal cycle of the STL ozone, which is high in spring and low in summer, it is inconsistent with the seasonal cycle of the TRL ozone, which is the highest in early

summer, thereby showing a relatively low correlation with TRL ozone. Moreover, the SOPL ozone column shows a positive correlation (at an r of approximately 0.3) with wind speed in the SOPL and upper troposphere (3–9 km), and a negative correlation with the relative humidity of the SOPL, corresponding to an r of −0.48. These characteristics illustrate the tendency of ozone in the SOPL to increase as the vertical mixing becomes stronger from strong winds. According to Mohanakumar [42], vertical mixing in the troposphere takes hours to days, while mixing in the stratosphere takes months to years. Furthermore, the vertical transport of air and chemical species through mid-latitude baroclinic eddies occurs at timescales of days. Therefore, it may take several days or more for the effect of vertical mixing due to wind to be reflected in atmospheric ozone concentrations, which may be a factor that weakens the correlation coefficient ($p$-value > 0.05) between wind and the ozone concentration in the SOPL. Among the wind components, the correlation with wind speed in the upper troposphere (WS_UTR in Figure 4) is higher than that of the SOPL, which is thought to be the result of strong wind in the lower layer enhancing vertical mixing [43–45]. Contrarily, the TRL ozone column showed the strongest correlation with the surface ozone, while it showed a negative correlation with the wind components. Although, under certain conditions, wind plays an important role in generating high concentrations of ozone in the lower layers [46–49], the correlation between the time series is shown to be low due to the opposite seasonality shown by the TRL ozone and the wind components. Meanwhile, the correlation heatmap in summer shows a different pattern. Almost all components have a high correlation with changes in the ozone concentration of the SOPL and TRL. Ozone in both the SOPL and TRL shows positive correlations with wind components as well as surface ozone, while it was negatively correlated with relative humidity. In this period, the atmosphere is considerably unstable compared to the other seasons, and wind components serve to further accelerate the vertical mixing of the atmosphere [50]. Air in the troposphere, containing high concentrations of ozone owing to photochemical reactions, is continuously mixed to affect the SOPL, which has similar characteristics to the TRL as the tropopause increases. That is, the influence of the wind component, which is limited due to the relatively weak vertical mixing in spring, is amplified due to the unstable atmosphere in summer.

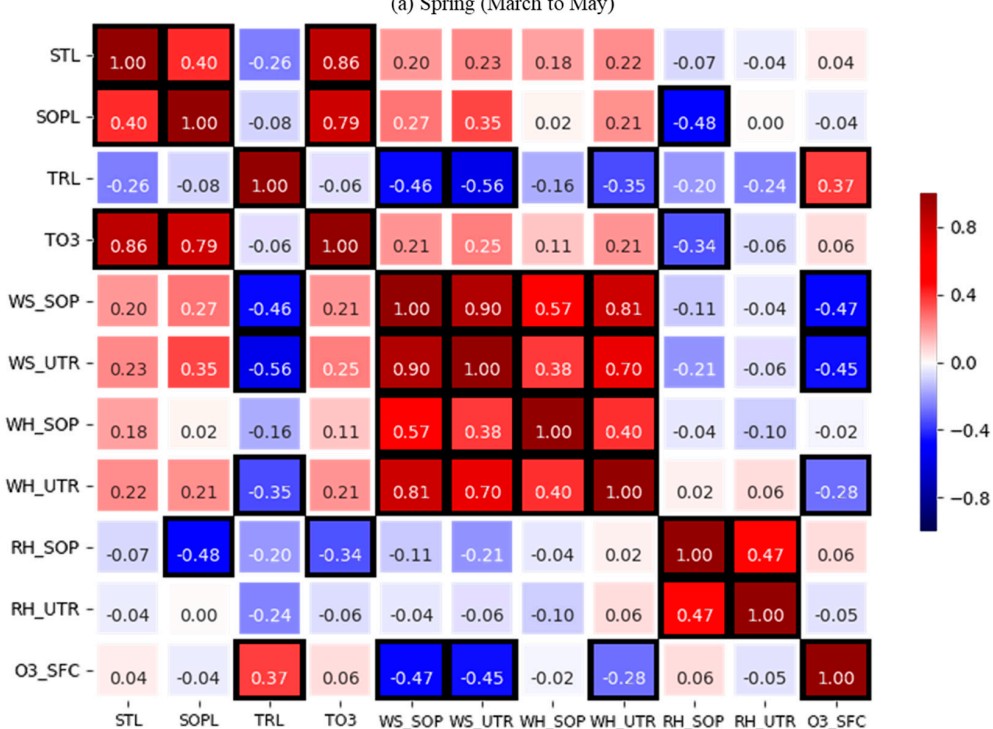

**Figure 4.** *Cont.*

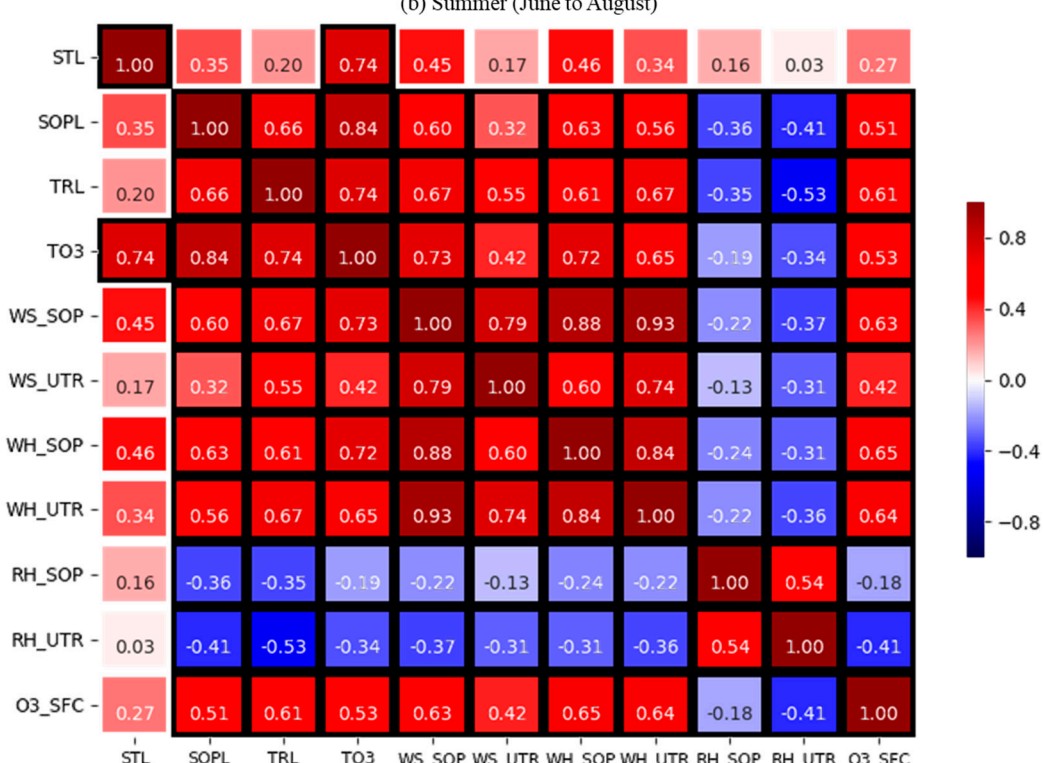

**Figure 4.** Correlation heatmap of each component (STL: stratospheric layer, SOPL: second ozone peak layer, TRL: tropospheric layer, TO3: total ozone, WS_SOP: wind speed of SOPL, WS_UTR: wind speed of the upper troposphere (3–9 km), WH_SOP: wind shear of SOPL, WH_UTR: wind shear of upper troposphere, RH_SOP: relative humidity of SOPL, RH_UTR: relative humidity of the upper troposphere, O3_SFC: surface ozone) for (**a**) spring, and (**b**) summer seasons. The black boundary line of each box represents the correlation values with a *p*-value < 0.05.

Figure 5 shows the results of the trend analysis of the components that showed strong correlations with ozone variations in the SOPL and TRL (wind speed of upper troposphere and relative humidity of the SOPL, surface ozone) for each season. The rate of change of each component is represented by a percentile deviation with the seasonality removed, as shown in Figure 3. The wind speed of the upper troposphere has the highest increase of 4.15% decade$^{-1}$ in fall, while that in the other seasons similarly increase at around 2% decade$^{-1}$. The relative humidity shows the weakest decrease in summer and a large decrease approaching about 50% decade$^{-1}$ in the other seasons, indicating that the inflow of dry upper air in the SOPL is increasing. Additionally, the decreasing trend of relative humidity in summer indicates the possibility that dry upper air can also affect the TRL. However, according to previous studies, the variation in the concentration of tropospheric ozone is predominately affected by ozone generated at the surface [23,41], and the surface ozone of Pohang shown in Figure 5c shows a strong increasing trend of around 20% decade$^{-1}$ in spring and summer. Therefore, the long-term increase in ozone in the TRL is thought to be the result of the steady increase of surface ozone due to industrialization in East Asia, along with the influx of air from the STL.

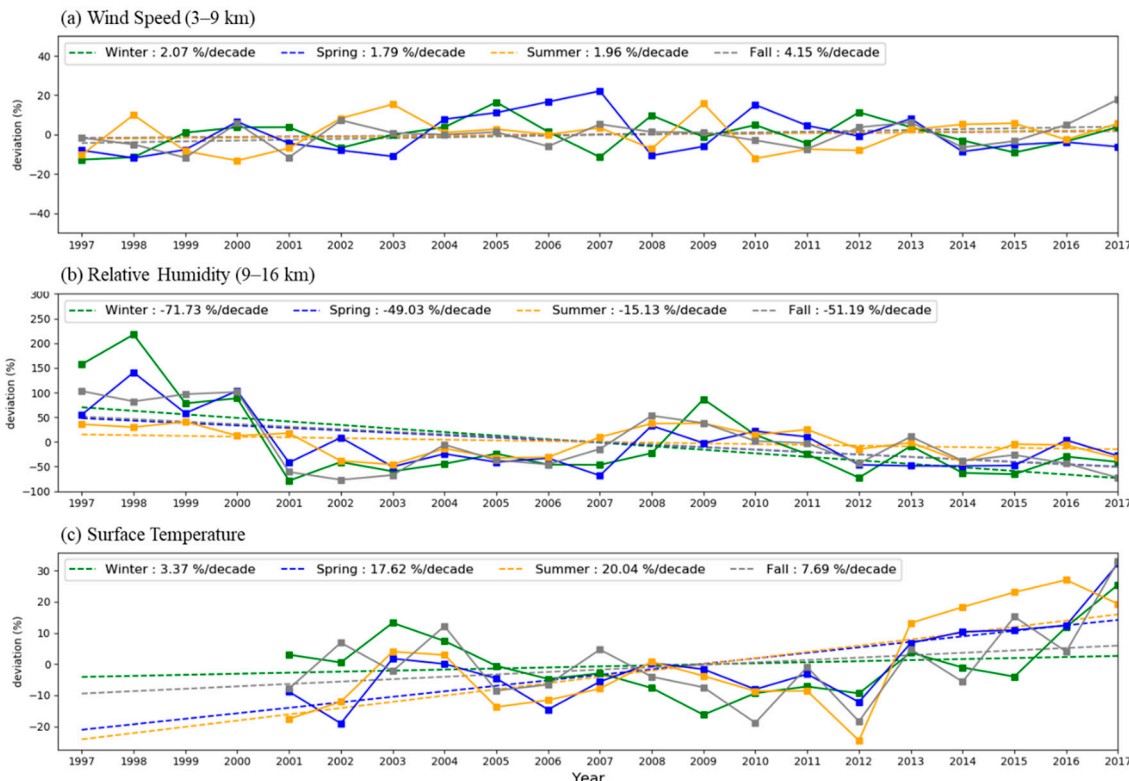

**Figure 5.** Time series of percent deviations in the de-seasonalized average ozone (DU) of (**a**) wind speed of the upper troposphere (3–9 km), (**b**) humidity of the SOPL, and (**c**) surface ozone for each season. The solid and dashed lines indicate the annual mean and trend, respectively (green: winter, blue: spring, yellow: summer, gray: fall).

### 3.3. Characteristics of the Second Ozone Peak Generation

Meteorological characteristics related to the high-level ozone in the SOPL can be identified through the following case analysis. The second ozone peak is closely related to the upper jet stream in terms of synoptic scale [51], and on a global scale, it is also connected to the springtime downwelling of air masses in the mid-latitudes of the northern hemisphere (the Brewer–Dobson circulation) [52]. Strong horizontal wind is accompanied by vertical wind shear, leading to tropopause folds and the inflow of high concentrations of ozone into the upper layer of the troposphere [53]. Based on the 21-year vertical ozone partial pressure observed with an ozonesonde, the second ozone peak appears mainly at a 10–14 km altitude in winter and spring (Figure 6a), and the strongest jet stream occurs at a 12–15 km altitude, which can be identified in vertical winds observed with radiosondes (Figure 6b). The ozonesonde observations of the cases with high-level ozone in the SOPL are shown in Figure 7. In all cases, the ozone columns of the SOPL were over 50% higher than the average for the same period. The first (19 May 2009) and second (2 May 2014) case are in the springtime when the STE is most active, which show an obvious second ozone peak at approximately 9–16 km. Yet, the last case (22 August 2014) is in the summertime and has no distinct second ozone peak. In the first case, the wind speed of the SOPL is significantly higher than the mean value, and in the second case, the wind speed is only slightly higher than the average at the altitude just below the SOPL. In the third case, the wind speed is strong and the difference in the vertical wind speed is significantly larger compared to the average of the same period. Weather maps of the case days (Figure 8) for the first and second cases indicate that the Pohang area is affected by strong wind advection along the main axis of the jet stream at 200 hPa (Figure 8a). The cut-off low is on the east side trough located on the Korean Peninsula at 500 hPa (Figure 8b). Note that the ground is under the influence of high pressure (Figure 8c). The backward trajectory analysis passing through Pohang (Figure 9) shows where the air

mass containing high ozone concentrations is introduced on the case days. The air masses in the SOPL (at approximately 12 km) in the first case come from higher latitudes and a similar altitude, and the air in the second case originates in an area more biased toward the polar region and passes through a very high ozone area over 380 DU. The pattern of the air origin shifting north with respect to the second ozone peak is consistent with that mentioned in a previous study [14]. The closer to the pole, the lower the tropopause, so that air masses in higher latitude are likely to contain high concentrations of ozone. In particular, a strong gradient of ozone concentrations is located near the mid-latitude region in spring, and, therefore, air from high latitudes may contain a much higher level of ozone than the surroundings, which can explain the formation of a strong second ozone peak in the second case, despite the fact that the wind speed is not significantly stronger than the average value. In conclusion, the first case appears to be influenced by high ozone concentrations at high latitudes and regional mixing due to strong winds, and the second case seems to be primarily influenced by high ozone concentrations at high latitudes rather than strong winds. In the third case, air masses mainly move in the longitudinal direction, in contrast to those in the other cases. In addition, the contribution of lower altitude air is relatively high, which can be seen through the penetration of air in the lower layer with high relative humidity into the SOPL (Figure 7). That is, high ozone concentrations in the lower layer affect the SOPL ozone in summer due to active vertical mixing. The case analysis indicates that there is a distinct difference in the mechanisms generating high ozone concentrations in the SOPL in summer and spring, and it is necessary to distinguish between them.

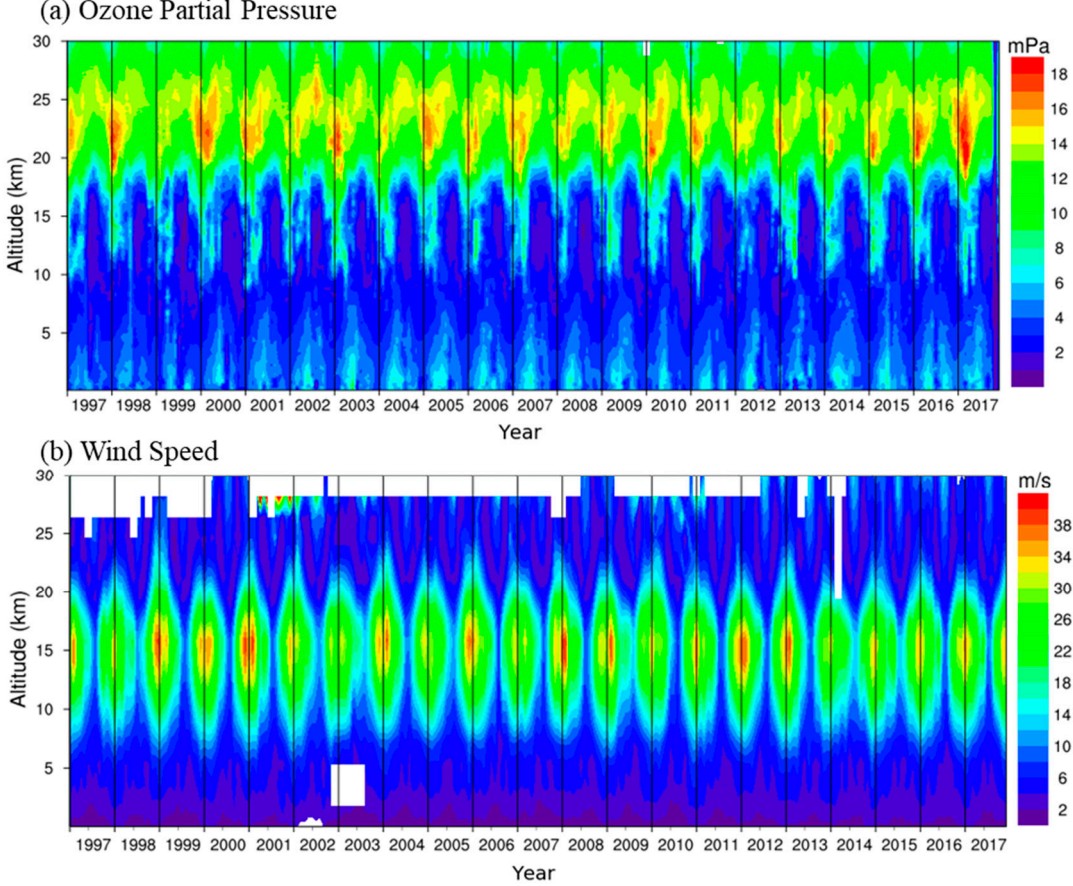

**Figure 6.** Monthly variation of time–altitude cross section of (**a**) ozone partial pressure and (**b**) wind speed at Pohang for 20 years (from 1997 to 2017).

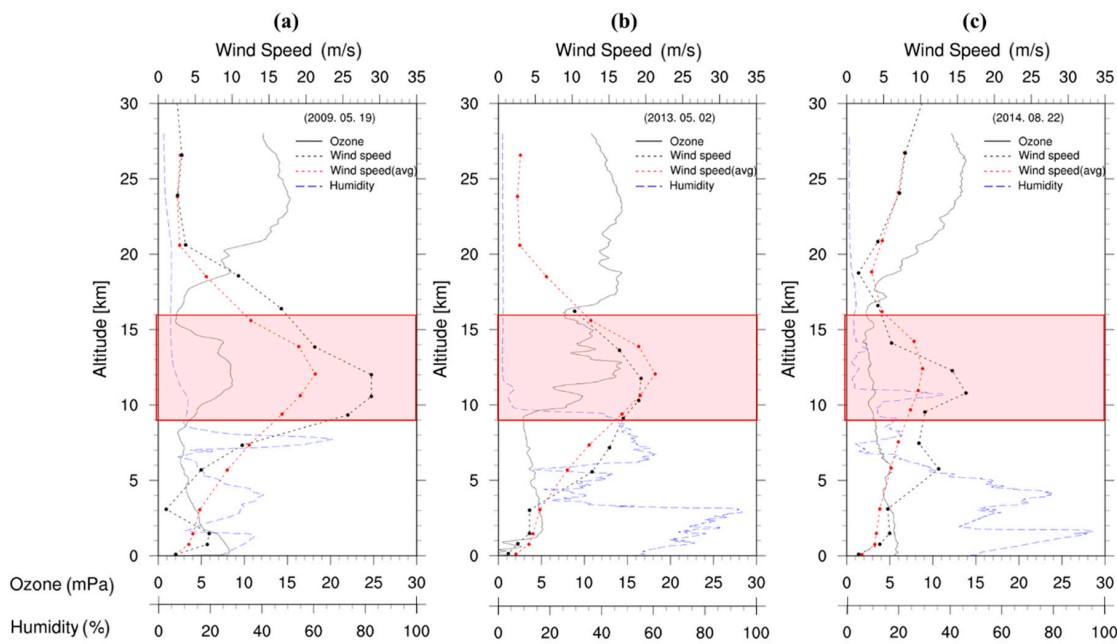

**Figure 7.** Vertical profile of ozone partial pressure and air temperature on (**a**) 19 May 2009, (**b**) 2 May 2013, and (**c**) 22 August 2014, respectively. The shaded area represents the second ozone peak layer.

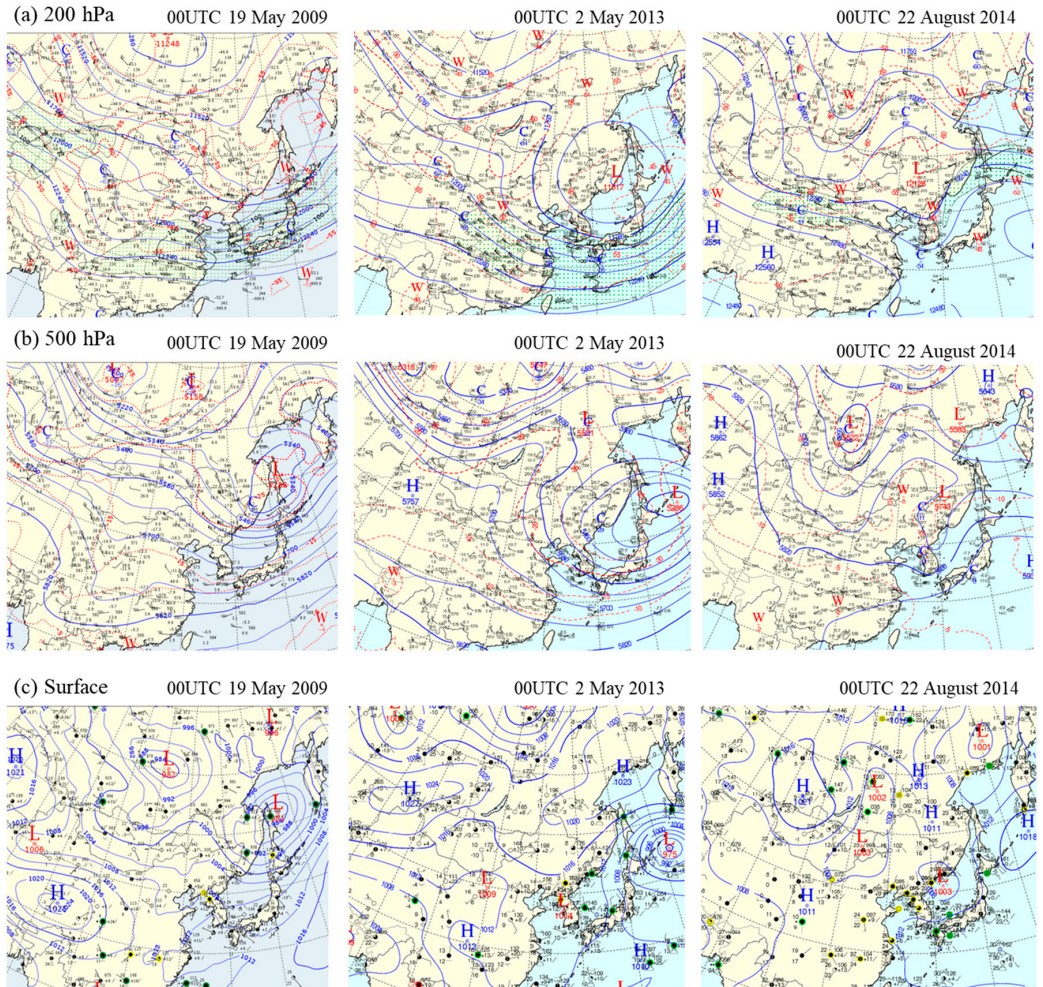

**Figure 8.** Synoptic weather maps of (**a**) 200 hPa, (**b**) 500 hPa, and (**c**) surface at 00:00 UTC 19 May 2009, 2 May 2013, and 22 August 2014, respectively.

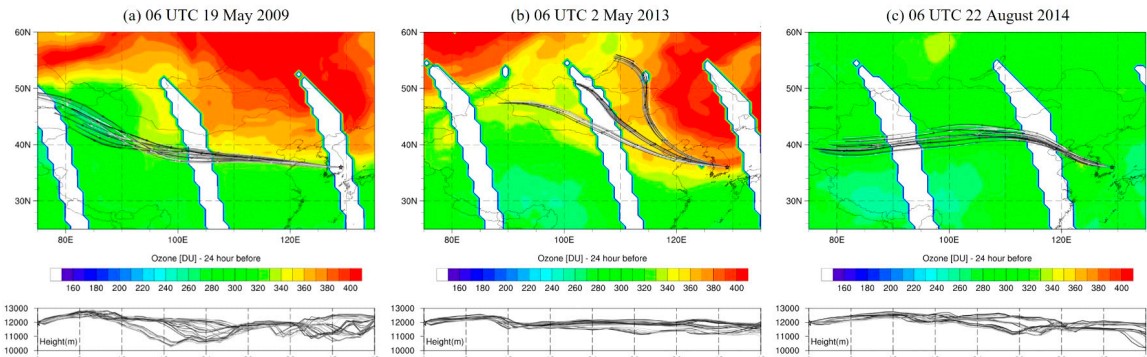

**Figure 9.** Results of 48 h backward trajectory over Pohang on (**a**) 19 May 2009, (**b**) 2 May 2013, and (**c**) 22 August 2014, respectively. The background contour represents the total column ozone of the day before each case obtained from the Ozone Monitoring Instrument (OMI)/ Aura satellite data (Level 3).

## 4. Conclusions

In this study, the ozone variations in the mid-latitude region of the northern hemisphere were examined through the vertical ozone observation data of Pohang after the regulation of CFC-11 emissions was enforced by the Montreal Protocol. The observed vertical ozone was divided into three layers—the STL, SOPL, and the TRL—and variations in the ozone concentrations of these layers were observed from 1997 to 2017. Our results show that during the entire study period, increasing ozone concentration trends of 0.45%, 2.64%, and 5.26% decade$^{-1}$ were observed for the STL, SOPL, and TRL, respectively. The ozone variations observed in the SOPL during springtime can be primarily attributed to an acceleration of STE, considering the long-term variation trend of the wind speed in the upper troposphere and relative humidity in the SOPL (Figure 5), in addition to the profiles of relative humidity in high ozone SOPL cases (Figure 7). Yet, transport from the polluted area may have some impact as the vertical movement of the atmosphere begins. However, during summer, the significant increase in ozone concentrations in the SOPL seems to be the result of increased locally generated ozone and transported pollutants from surrounding areas because it is mainly dominated by air from lower layers. The noticeable increase observed in the TRL ozone is considered to be significantly affected by surface ozone, which shows a considerable increase of over 20% decade$^{-1}$ in summer, in which the influx of stratospheric air may have some impact. Considering the strong increase in summer, it can be assumed that the ozone concentrations in the TRL are significantly affected by industrialization in the East Asia region. In addition, for the increase in STL during 2011 to 2016, the recent increase of CFC-11 emissions is a possible explanation, but further study is needed on this topic. The Korean Peninsula is located in a sphere of influence of China, which has undergone rapid industrialization, and Siberia, where massive wildfires occur every year [39,41,54]. It is estimated that the polluted air in these surrounding areas mainly affect the TRL and further expands in summer, with effects reaching the SOPL, thus contributing to the high growth rate of the TRL and SOPL in summer.

In the past, ozone variation analyses were mainly focused on stratospheric ozone due to the occurrence of ozone holes in this layer. However, in recent years, there has been a continuous increase in ozone in the SOPL, where STE occurs, and in the troposphere. Therefore, it has become crucial to monitor tropospheric ozone due to the increase in precursors caused by climate change and industrialization. Since high-level ozone on the ground has adverse effects on the human body, it is necessary to continuously monitor and take appropriate measures for the variations in tropospheric ozone. In response to these demands, the Korean Ministry of Environment launched a geostationary multi-purpose satellite GEO-KOMPSAT-2B (GK-2B) equipped with an environmental payload in February 2020 to continuously monitor air pollutants such as ozone, nitrogen oxides, and aerosols. A geostationary satellite can perform extensive analyses due to its high time resolution of several hours, and it is expected to improve the monitoring and analysis of ozone variations. In addition, efforts to

mitigate the negative effects of ozone in advance need to be an on-going process, made possible by more diverse and in-depth studies using numerical modeling techniques.

**Author Contributions:** Conceiving and designing the experiments, S.-B.R. and S.-S.L.; writing—original draft preparation, D.S. and S.S.; writing—review and editing, S.-B.R. and S.-S.L. All authors have read and agreed to the published version of the manuscript.

**Funding:** This work was funded by the Korea Meteorological Administration Research and Development Program "Development of Monitoring and Analysis Techniques for Atmospheric Composition in Korea" under Grant (KMA2018-00522).

**Acknowledgments:** The OMI/Aura TOMS-Like Ozone L3 data product was obtained from the Goddard Data and Information Services Center (DISC): http://disc.sci.gsfc.nasa.gov/Aura.

**Conflicts of Interest:** The authors declare no conflict of interest.

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
