# Peer review of "Variations in Ozone Concentration over the Mid-Latitude Region Revealed by Ozonesonde Observations in Pohang, South Korea"

_atmosphere, doi:10.3390/atmos11070746_

Round 1

Reviewer 1 Report

The manuscript “Variations in ozone concentration over the mid-latitude region revealed by ozonesonde observations 4 in Pohang, South Korea” presents an interesting analysis on the temporal variation of vertical ozone profiles in South Korea over a period of 22 years. The quality of the paper is descent and may be considered for publication however, there are a few analytical flaws and omissions to be resolved prior to publication.

The analytical flaw, that appears to be easily flexible, relates to the decadal estimates of changes in ozone concentrations. The regression is not described in detailed in the Methods (whether event, monthly or annual ozone concentrations were used for the regression analysis. I can only deduct from Figure 1 that the regression used annual ozone concentrations. In this case, the slope of the regression corresponds to the change in ozone concentration by year. I assume that, changes of ozone concentration per decade was most likely calculated by multiplying the slope by ten but this is not accurate, mathematically (since only two decades of data were available) and given the variability of ozone concentration (interannual trend was highly variable). To rectify it, authors should clearly describe how regression analysis was done and estimates were obtained. If my hypothesis above is correct, authors should report annual trends rather than decadal trends.

The omission relates to the contribution of ozone in the upper troposphere from distant sources. Authors did not consider this throughout their manuscript. The manuscript appeared to be centered on the penetration of stratospheric ozone. In the case of South Korea, it is known that it is downwind of major sources of anthropogenic pollution in China that can result in the formation of ozone aloft and transport. Moreover, there is evidence of the potential impacts of wildfires in Siberia and the formation of ozone aloft. In the case of regional aloft ozone it is reasonable to expect that high wind shear/speed will be associated with high aloft ozone concentrations. In general, it is understood that stratospheric ozone penetrations are rare in occurrence and when they occur, they result in substantial increases of ozone mixing ratios (in the order of ppm). It is recommended that authors should comprehensively test the potential for aloft regional ozone by analyzing the trajectory of air masses (using the meteorological data already available to them) in relation to anthropogenic sources in China and location of wildfires in Siberia. This is particularly important in order to conclude that stratospheric ozone penetration is driving upper troposphere ozone changes.

The current version of the manuscript needs editing however, it may be extensively revised to address the aforementioned technical flaw and omission. In general it is suggested that authors are consistent in their language and format of figures and tables. For example in figure 6 the scales of Y- and X-axis are different and the relationship is not obvious since plots (a) and (b) can be viewed concurrently to understand the association.  

Reviewer 2 Report

The paper is very well written and the literature review is extensive and recent.

The methods are sound and clearly explained. The results are clearly presented and summarized in the conclusions.

However, before it is considered for publication I suggest the authors to consider the following remarks:

Trends and correlations must be presented with the respective statistical significance.

I notice that some graphs in figures 1 and 5 show also clear periodic variability, perhaps more significant that the long-term trend. This means that the trend has some variability within the period if it is calculated for sub-periods. Please comment on this.

Round 2

Reviewer 1 Report

No comments

Reviewer 2 Report

The authors have satisfactorily addressed the raised issues.